# Patient-Reported Reasons for Not Using Primary Prevention Statin Therapy

**DOI:** 10.3390/jcm9103337

**Published:** 2020-10-18

**Authors:** Cassandra M. DeWitt, Robert B. Ponce, Hayley Bry, Soma Wali, Erica Sedlander, Joseph A. Ladapo

**Affiliations:** 1Division of General Internal Medicine and Health Services Research, David Geffen School of Medicine at UCLA, Los Angeles, CA 90024, USA; cdewitt@mednet.ucla.edu; 2College of Medicine, University of Cincinnati, Cincinnati, OH 45267, USA; rbponce@uci.edu; 3University of Southern California, Los Angeles, CA 90007, USA; Hbry@usc.edu; 4Department of Medicine, Olive View-UCLA Medical Center, Sylmar, CA 91342, USA; swali@dhs.lacounty.gov; 5Milken Institute School of Public Health, The George Washington University, Washington, DC 20052, USA; esedlander@gwu.edu

**Keywords:** statin, underuse, cardiovascular risk, cholesterol

## Abstract

Almost half of patients who meet American College of Cardiology/American Heart Association (ACC/AHA) criteria for statin therapy are untreated. We aimed to evaluate patient-reported reasons for not using guideline-recommended statin therapy in a public healthcare system. Achieving this goal is key to addressing gaps in care and reducing preventable cardiovascular morbidity. We surveyed patients who met 2013 ACC/AHA guidelines for statin therapy but were not using statins. The survey probed domains of patient knowledge regarding cardiovascular health and benefits of statins, barriers to use, physician trust, and interest in cardiovascular care. Among 71 patients eligible for guideline-recommended statin therapy but not currently taking statins, 49 (69%) had a high school education or lower, 41 (58%) reported that they were unaware they should be prescribed a statin and 49 (69%) were unaware of the benefits of statins. Almost all patients, 70 (99%), reported caring about their cardiovascular health, 61 (86%) reported that they had a high level of trust in their physician, and 51 (72%) reported a willingness to follow their physician’s advice. Despite interest in cardiovascular health, awareness of benefits of statin therapy was low and knowledge of recommended statin therapy was low. Increasing patients’ awareness of their eligibility through systematic testing and linkage to statin therapy, along with education, may increase statin use among patients recommended for therapy.

## 1. Introduction

Statin therapy reduces the risk of major adverse cardiovascular events. The 2013 American Heart Association/American College of Cardiology (AHA/ACC) guidelines recommend statin therapy for adults with clinical atherosclerotic cardiovascular disease (ASCVD), LDL-C levels of 190 mg per dL or greater, a diagnosis of diabetes mellitus, and LDL-C levels of 70 mg per dL or greater, or a 10-year ASCVD risk of 7.5% or greater. These guidelines increased the proportion of United States (U.S.) adults recommended for therapy from 43.2 million (35.5%) to 56 million (48.6%) [1,2,3,4]. However, a substantial proportion (44.5%) of patients who are eligible for guideline-recommended statin therapy do not use statins.[5] This is particularly a problem for racial/ethnic minorities and uninsured patients, who are less likely to use statins compared to white patients and patients with private insurance, respectively [6].

In a prior study of 10,138 participants, patients who discontinued statins were more likely than patients who continued statins to report that their physicians had not adequately explained the significance of their cholesterol levels and how dyslipidemia contributes to cardiovascular risk [7]. A recent study found that women, black adults, and those without insurance were more likely to report never being offered a statin despite being eligible for a statin [8]. In examining differences between those who continued versus those who discontinued statin therapy, members of the former group were more likely to have follow-up and laboratory work within six months of initiating statins [9]. Furthermore, they were more likely to trust their physicians than patients who discontinued statin therapy [9]. Patients in focus groups from a Kaiser Permanente Northern California study reported a variety of reasons for discontinuing statins, including uncertainty about the importance or benefits of statins, concerns about side effects, and inconvenience associated with taking medications [10]. Another study of 1220 participants who previously used statins found that 60% cited muscle pain as the main reason for discontinuation, while 16% reported cost as a barrier to adherence [11]. While most prior research has focused on patients who discontinued statin therapy, less is known about why patients may never have even been initiated on statin therapy.

As part of an effort to increase the use of guideline-recommended statin therapy for primary prevention in a public health system with a low-income population, we administered a survey to patients who met 2013 ACC/AHA guidelines for statin therapy but were not using statins. We aimed to evaluate patient-reported reasons for not using guideline-recommended statin therapy. The survey addressed patient knowledge regarding cardiovascular health and benefits of statins, barriers to use, physician trust, and patient interest in cardiovascular care. We hypothesized that a substantial proportion of patients would be unaware that they were appropriate candidates for statin therapy and unaware that statin therapy would reduce their risk of cardiovascular disease.

## 2. Experimental Section

We surveyed patients receiving care at Olive View-UCLA Medical Center who met 2013 ACC/AHA criteria for statin therapy but were not using statins. Olive View-UCLA Medical Center is part of the Los Angeles County Department of Health Services (LAC DHS) and serves a large proportion of the LAC’s low-income population. Patients were identified using the electronic health record (EHR) and randomly selected for contact, and surveys were conducted by phone between 1 February 2018 and 31 January 2019. Patients who completed the survey were mailed a $10 department store gift card for compensation.

Patients were eligible for participation if they spoke English or Spanish, were age 18–70 years-old, had at least one outpatient primary care visit at Olive View-UCLA Medical Center within the preceding two years, met 2013 ACC/AHA criteria for statin therapy, but were not using statins, and were able to provide informed consent. Our sample was limited to patients without established ASCVD, whose eligibility for statin therapy was therefore based on primary prevention as informed by the 10-year ASCVD risk. We further limited the sample to patients with obesity (body mass index, BMI ≥ 30) because the survey study was partially supported by funding targeting obesity. Patients were contacted based on eligibility criteria available in the EHR, including preferred language, age, visit history, ASCVD score/cardiovascular risk factors, and medication history. We did not exclude patients using other lipid-lowering supplements or drugs, such as fibrates or ezetimibe, because our study focus was statin therapy. The absence of current statin use was confirmed during the phone interview. We contacted patients from approximately March 2018 to June 2019 and the electronic health records were used to determine eligibility extended to 2016. We used data available at the most recent encounter to identify statin use and did not assess for prior statin use.

We developed survey questions to address four domains that we hypothesized may contribute to the absence of statin therapy in patients who meet 2013 ACC/AHA guidelines for treatment. Domains included lack of patient knowledge regarding cardiovascular health and benefits of statins, economic/social barriers to statin use, low physician trust, and low interest in cardiovascular care/health. In addition to these questions, the survey also included a physician trust questionnaire and a medication adherence question adapted from prior studies [12,13,14]. Survey development was also informed by prior work which assessed patient understanding of results of cardiac stress testing and demonstrated that increased knowledge and understanding were associated with less frequent subsequent cardiovascular testing [15]. The physician trust question was, “All things considered, how much do you trust your doctor (Scale of 0 to 10; 0 = Not at all and 10 = Completely)?” Adapting from prior work using a physician trust scale, we considered a score of 8 or higher to signify a high level of trust [16]. All questions were written at the sixth-grade reading level or below. The survey was translated into Spanish by a certified Spanish interpreter.

The primary survey measure was the proportion of patients not currently receiving guideline-recommended statin therapy who reported being unaware that they were candidates for treatment (as determined by their response to the question, “Have any of your doctors ever told you that you should be taking a statin to lower your cholesterol?”). We report summary statistics and percentages for survey measures. We estimated a logistic regression model to evaluate the relationship between socioeconomic characteristics and our primary measure. This model included sex, age 65 or older, college education (some college or college graduate), travel time of 30 min or more to visit a doctor, and a history of not taking medication because of cost. We report adjusted odds ratios and 95% confidence intervals (CIs). All analyses were performed using Stata, version 14 (StataCorp, Inc., College Station, TX, USA).

## 3. Results

We contacted 116 patients and 71 met all eligibility criteria and completed the survey. The sample included 25 (35%) women, mean age was 62 years, 51 (72%) were Hispanic, 49 (69%) reported having a high school education or lower, and 57 (80%) reported never using a statin in the past. The mean 10-year ASCVD score was 19%. Other patient characteristics are shown in Table 1.

Half of the participants, 36 (51%) reported their provider had never told them that their cholesterol was elevated, 41 (58%) reported never being told by a doctor to use a statin to lower their cholesterol, and 49 (69%) reported never being told that a statin would lower their cardiovascular risk. While nearly all participants, 66 (93%) reported having a doctor they could see when needed for illness, 26 (37%) participants reported traveling at least 30 min for a visit, and 18 (25%) participants reported not taking a medication due to cost. Other findings regarding knowledge about statin therapy and barriers to care are shown in Table 2.

Almost all participants reported an interest in their cardiovascular health, 70 (99%) participants, reported that they agreed or strongly agreed with a statement about caring about “heart health” and 43 (61%) reported an interest in taking a statin for cardiovascular benefits. Fifty-one participants (72%) reported always or almost always getting follow-up care recommended by a doctor. The mean physician trust score was 8.84 (SD = 1.5) (on a scale from 0 to 10, with 10 being complete trust). Other findings regarding patient interest in cardiovascular health and follow up care can be found in Table 3.

In our regression model evaluating the relationship between socioeconomic characteristics and our primary measure, there was no significant association with female sex (adjusted odds ratio [aOR] 0.77; 95% CI, 0.26–2.28; *p* = 0.64), age 65 and older (aOR 0.91; 95% CI, 0.29–2.86; *p* = 0.86), college education (aOR 0.99; 95% CI, 0.31–3.10; *p* = 0.98), or not taking medications due to cost (aOR 0.57; 95% CI, 0.17–1.92; *p* = 0.37). Having a travel time to see a doctor of 30 min or more was associated with a lower likelihood of reporting that the doctor did not advise the patient to take a statin for cholesterol (aOR 0.28; 95% CI, 0.09–0.86; *p* = 0.03).

## 4. Discussion

We found that half of the patients not receiving guideline-recommended statin therapy for primary prevention were unaware that they were candidates for therapy, and 49 (69%) were unaware that a statin could lower their risk of having an adverse cardiovascular event or dying. In this cohort of patients receiving care in the Los Angeles County public health system, socioeconomic barriers to therapy were mixed, with most patients having access to a primary care doctor, but many reported long travel times to see their physician and a history of not taking medication because of cost. In addition, most patients reported an interest in cardiovascular health and almost two-thirds of patients reported being willing to take a statin. Our regression analyses demonstrated an inverse relationship between longer travel times to see a doctor and the likelihood of a patient reporting to have never been advised to take a statin. This finding may be due to selection bias if patients were more willing to travel further for higher quality doctors, and higher quality doctors were more likely to advise their patients to use guideline-recommended medication.

Statin therapy is underused in the United States among patients who meet guideline recommendations for primary and secondary prevention [17,18,19]. Underuse is particularly pronounced for uninsured patients and racial/ethnic minorities, who are substantially less likely to be prescribed statin therapy compared to patients with insurance or who are white [6,8,17]. These disparities in care have persisted despite wide adoption of generic statins, which now comprise 82% of statins prescribed in the US [6]. Our survey participants were almost entirely racial/ethnic minorities, and based on the population characteristics of patients receiving care in LAC DHS, a substantial proportion was uninsured. This cohort is therefore at markedly high risk of not receiving guideline-recommended statin therapy [6,8,17].

Our survey results provide support for tailored approaches to reducing the underuse of statin therapy among racial/ethnic minorities. While nearly all participants reported having undergone cholesterol testing, fewer than half of participants reported being told by a physician that they would benefit from statin therapy, or that statin therapy would lower their risk of an adverse cardiovascular event. This suggests a critical missing link in optimizing preventive cardiovascular care for primary prevention occurs after lipid testing. Patients may not have been seen in follow-up to review lipid testing results, or continuity of care may have been disrupted for patient- or provider-related reasons. This challenge may be ameliorated by pre-emptively discussing cardiovascular risk and statin therapy at the time of testing. This strategy could be coupled with pre-emptive or automated prescribing of a statin, with a prescription automatically triggered pending the results of lipid testing, for example. This type of model may also be beneficial to patients who travel long distances to see their physicians. Moreover, because rates of provider trust were high, provider-based improvements in communication may yield substantial improvements in statin use rates. Prior studies show that multilevel strategies targeting physician prescribing patterns and beliefs about the safety of statin therapy may also be beneficial [20].

While monthly out-of-pocket costs for medications were moderate in this cohort, these costs represent a higher proportion of overall income in the LAC DHS population compared to patients with private insurance. There is evidence that further reductions in cost sharing for cardiovascular medications may also improve adherence and health outcomes [21]. The wide availability of effective generic statins may further attenuate economic concerns as a barrier to statin therapy.

Our study has several limitations. We were unable to determine whether the absence of guideline-recommended statin therapy in patients was attributable to patient-factors, such as loss to follow-up or misconceptions about the risk of statin therapy, or attributable to physician-factors, such as provider prioritization of more urgent clinical issues, or not adequately addressing patient concerns or fears about statin therapy. Our study population was small and this limits inferences on generalized epidemiologic relationships. In addition, challenges related to access to care or medications, prevalent community beliefs about cardiovascular risk factors, and socioeconomic differences may also constrain our ability to make general inferences. Further, because the population was limited to patients with obesity receiving care in LAC DHS, and their barriers to guideline-recommended statin therapy may differ from other populations, including populations with a markedly different racial/ethnic composition. This study focused on patient-centered barriers to care, but provider-centered barriers may also contribute to lower rates of guideline-recommended care. Provider-centered interventions would be another possible target of interventions to increase guideline-concordant care [22].

Despite a high level of interest in cardiovascular health among patients, awareness of the benefits of statin therapy is low and knowledge of eligibility for guideline-recommended statin treatment is also low. Increasing patients’ awareness of their eligibility through systematic testing and linkage to statin therapy, along with patient education, may increase statin use among patients recommended for therapy.

## Figures and Tables

**Table 1 jcm-09-03337-t001:** Demographic Characteristics of Participants.

Characteristic	Number of Participants (*n* = 71)	Percentage
**Mean Age (year)**	62.3	
**Male Sex (%)**	46	65%
**Race**		
Hispanic or Latino Origin	51	72%
White, non-Hispanic	3	4%
Black, non-Hispanic	4	6%
Other or unknown	13	18%
**Education Level**		
High school or lower	49	69%
Some college	9	13%
College graduate	13	18%
**ASCVD ^1^ Risk Factors**		
Diabetic w/LDL-C ^2^ < 70	34	48%
LDL-C > 190	11	15%
ASCVD risk ≥ 7.5%	65	92%

^1^ ASCVD risk: Atherosclerotic Cardiovascular Disease; ^2^ LDL: Low Density Lipoprotein.

**Table 2 jcm-09-03337-t002:** Knowledge about Statin therapy and Barriers to Care.

Characteristic	Number of Participants (*n* = 71)	Percentage
**Knowledge and Awareness of Cardiovascular Disease and Statin Therapy**		
Have you ever had a blood test to check your cholesterol?		
Yes	70	99%
No/unsure	1	1%
Have you ever been told your cholesterol was high?		
Yes	35	49%
No/unsure	36	51%
Have any of your doctors ever told you that you should be taking a statin to lower your cholesterol?		
Yes	30	42%
No/unsure	41	58%
Have you ever been told a statin would lower your risk of having a heart attack, stroke or dying?		
Yes	22	31%
No/unsure	49	69%
**Socioeconomic Barriers to Statin Therapy**		
Do you have a primary care doctor or doctor that you see regularly?		
Yes	62	87%
No/unsure	9	13%
If you were to become sick is there a doctor with whom you can make an appointment?		
Yes	66	93%
No/unsure	5	7%
Travel time to visit a doctor		
<15 min	19	27%
16–30 min	26	37%
30–45 min	14	20%
>45 min	12	17%
Monthly out-of-pocket costs for medications		
$0–25	0	0%
$26–50	62	87%
$51–150	1	6%
>$150	3	7%
Have you ever not taken a medication because cost prevented you from getting the medication?		
Yes	18	25%
No/unsure	53	75%

**Table 3 jcm-09-03337-t003:** Patient Interest in Cardiovascular Health and Follow-up Care

Characteristic	Number of Participants (*n* = 71)	Percentage
**I care about my heart health**		
Strongly agree with statement	49	69%
Agree with statement	21	30%
Disagree with statement	0	0%
Strongly disagree with statement	0	0%
Missing	1	1%
**Would you be willing to take a statin to lower your risk of heart attack, stroke or dying?**		
Yes	43	61%
No/unsure	28	39%
**Are there any other medications, supplements or natural remedies you are using to lower your cholesterol?**		
Yes	12	17%
No/unsure	59	83%
**Adherence to follow-up care recommendations**		
Always or almost always get follow up care a doctor recommends	51	72%
Usually get care a doctor recommends	12	17%
Sometimes get care a doctor recommends	5	7%
Rarely get the follow up care a doctor recommends	3	4%

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
