# Peer review of "Patient-Reported Reasons for Not Using Primary Prevention Statin Therapy"

_jcm, 2020, doi:10.3390/jcm9103337_

Round 1
Reviewer 1 Report
ABSTRACT
- I suggest the author clearly specify what the aim of the paper is. This information should also be reported in the last part of the introduction.
INTRODUCTION
- To clarify the rationale and the context of the study, I think that it might be helpful to specify the 2013 American Heart Association/American College of Cardiology (AHA/ACC) criteria for the eligibility to statin treatments.
- I suggest the authors report the numbers of patients along with percentages throughout the text. Some examples ae reported below:
- Page 1, line 35: “from 43.2 million (37.5%) to 56 million (48.6%)”;
- Page 3, lines 102-103: “The sample included 25 women (35%), mean age 62 years, 51 (72%) were Hispanic, 49 (69%) reported having a high school education or lower, and 57 (80%) reported never using a statin in the past”.
METHODS AND RESULTS
- The primary survey measure was reported twice in the methods section (lines 69-71 and lines 95-98). To avoid redundancies, I suggest the authors to report the primary survey measure only once.
- In lines 75-77, the authors state that the study sample “was limited to patients without established atherosclerotic cardiovascular disease (ASCVD)”, but in the results section (lines 103-104) they report the proportion of included patients with ASCVD. Please clarify this. Moreover, in this regard, the authors state that “The mean ASCVD score was 19%”. I think that probably “mean score” is not the correct terminology to use in this case. Maybe “proportion of patients with ASCVD” is a more appropriate phrase to use. Please check this and modify accordingly.
- The authors state that patients not using statins were included in the study. What about patients treated with other lipid-lowering drugs (e.g. fibrates, ezetimibe or bile acid binding resins) or supplements? Were they excluded from the survey? Please clarify and specify this.
- Lines 66-67: The authors state that patients to be included in the survey were identified using electronic health records (EHR). Which timeframe was covered by the EHR? Was the entire available timeframe used to assess statin exposure (i.e. never treated patients) or was a washout period established? Please clarify this.
- To substantiate the results of the study, I would strongly suggest the authors carry out some sort of statistical analysis. For example, it could be interesting to investigate if the variables collected by the authors in the survey could be interpreted as non-treatment predictors.
Reviewer 2 Report
Patient-reported Reasons for Not Using Primary Prevention Statin Therapy
In the USA a high proportion of patients eligible for statin treatment (presumable more than 40%) do not use statins. Ethnic origin, lower social levels, education and lack of health insurance membership are regarded as some of the effectors determining this observation.
To better understand patient`s reported reasons for not using statins a small survey including 71 participants (mean age 62.3 y, 65% male and predominantly Hispanic origin) have been interviewed. Almost all participants had blood tests for cholesterol, but according to the participants remembrance only half of them was told of cholesterol being high. Similarly, the majority felt uniformed with respect to a potentially beneficial effect of statin treatment, despite almost all participants reported high interest in cardiovascular health, and there was a high level of trust in the treating physicians. The authors advocate to improve patient`s awareness about their cardiovascular risk and the potential benefits of primary prevention together with an intensified medical care.
General comments:
Although the data are of general interest and the conclusions at least in part are plausible, the cohort under investigation is too small and thereby including a high risk of selection bias. This cohort therefore rather may serve for a “wake-up call” but not as a solid basis for discussing generalized epidemiological relationships. Failure of statin treatment simply may be the result of not being a member of a health insurance, and this may be a local problem. Also the population-based knowledge on cardiovascular risk factors in general may severely influence the intensity of the individual health care as well as patient`s self-care. This also is well-known as a matter of education, social conditions and affiliation to different social classes.
Aiming a sound description and presentation of the problem under investigation, it would be more appropriate
(a) to perform a prospective or retrospective survey on a predefined population not taking statins compared with matched controls that appropriately take statins or
(b) to predefine geographical regions or social classes and then compare treatment adherence.
Another limitation is the inadequate presentation of patient`s medical history and medical findings. Moreover the patient`s reports have not been witnessed by checking medical records. In only 17% of the small cohort patients report to take – apart from statins – other medications. This appears to be questionable, as in the age of this cohort “multi-morbidity” is likely (hyperlipidemia, hypertension, diabetes).
In conclusion, I recommend to rearrange the manuscript to a “research letter” aiming a call for attention with respect to a potentially relevant lack of information, supervision and adherence to guideline recommended medical treatment, thereby also comparing different regions within the USA and different health care systems at least within the Western world.
Round 2
Reviewer 1 Report
The authors modified the manuscript according to the indications suggested. This is now a well-written and informative article. However, I have only a minor suggestion to make. As I have already stated in the previous revision, numbers and percentages could be reported more clearly, as follows: number (percentage); e.g.: The sample included 25 (35%) women , mean age 62 years, 51 (72%) were Hispanic, 49 (69%) reported having a high school education or lower, and 57 (80%) reported never using a statin in the past."
Reviewer 2 Report
Dear Dr. Lapado, dear Authors,
the manuscript has been improved. However, after regression analysis he major message remains that "Having a travel time to see a doctor of 30 minutes or more was associated with a lower likelihood of reporting that the doctor did not advise the patient to take a statin for cholesterol" . As a conclusion, this remains somewhat unsatisfactory. A "lack of information and support" can hardly be explained by the distance alone. Obviously, there is a general lack of communication between patients and their doctors. If this conclusion is correct, this potential problem should be more focussed, and some hypothetical reasons for this deficiency in medical information and support, as well as some potential ways to get aout of this problem may be added at the end of the discussion.
with kindest regards
Bernhard Rauch
